# Experimental protocol to investigate cortical, muscular and body representation alterations in adolescents with idiopathic scoliosis

**Matilde Paramento[1,2], Maria Rubega[1]\*, Roberto Di Marco[3], Paola Contessa[4], Michela Agostini[1], Francesca Cantele[1], Stefano Masiero[1,4,5], Emanuela Formaggio[1]**

**1** Department of Neurosciences, Section of Rehabilitation, University of Padova, Padova, Italy, **2** Department of Information Engineering, University of Padova, Padova, Italy, **3** Department of Engineering for Innovation Medicine, University of Verona, Verona, Italy, **4** Orthopedic Rehabilitation Unit, Padova University Hospital, Padova, Italy, **5** Ospedale Riabilitativo di Alta Specializzazione di Motta di Livenza, Motta di Livenza, Treviso, Italy

\* maria.rubega@unipd.it

## Abstract

### Background

Adolescent idiopathic scoliosis (AIS) is the most common form of scoliosis. AIS is a three-dimensional morphological spinal deformity that affects approximately 1-3% of adolescents. Not all factors related to the etiology of AIS have yet been identified.

### Objective

The primary aim of this experimental protocol is to quantitatively investigate alterations in body representation in AIS, and to quantitatively and objectively track the changes in body sensorimotor representation due to treatment.

### Methods

Adolescent girls with a confirmed diagnosis of mild (Cobb angle: 10˚-20˚) or moderate (21˚-35˚) scoliosis as well as age and sex-matched controls will be recruited. Participants will be asked to perform a 6-min upright standing and two tasks—named target reaching and fore-arm bisection task. Eventually, subjects will fill in a self-report questionnaire and a computer-based test to assess body image. This evaluation will be repeated after 6 and 12 months of treatment (i.e., partial or full-time brace and physiotherapy corrective postural exercises).

### Results

We expect that theta brain rhythm in the central brain areas, alpha brain rhythm lateralization and body representation will change over time depending on treatment and scoliosis progression as a compensatory strategy to overcome a sensorimotor dysfunction. We also expect asymmetric activation of the trunk muscle during reaching tasks and decreased postural stability in AIS.

relevant data from this study will be made available upon study completion.

**Funding:** The author(s) received no specific funding for this work.

**Competing interests:** The authors have declared that no competing interests exist.

## Conclusions

Quantitatively assess the body representation at different time points during AIS treatment may provide new insights on the pathophysiology and etiology of scoliosis.

## Introduction

Adolescent idiopathic scoliosis (AIS) is the most common form of scoliosis, affecting approximately 1–3% of adolescents (10–18 years of age) [1]. AIS is a three-dimensional morphological spinal deformity that typically manifests a translation and rotation of the spine in the frontal and horizontal planes, and most of the time a reduction in the physiological curves of the rachis in the sagittal plane. AIS significantly changes the structure and appearance of the adolescent spine, thorax, and trunk [2, 3]. The severity of scoliosis is typically assessed through the Cobb angle, which is the angle between the endplates of the upper and lower end vertebrae—the vertebrae with the largest angle to the horizontal—measured from X-rays. This parameter is considered the standard method to quantify the severity of scoliosis as well as to evaluate the need for treatment, monitor the progression of the curve, and estimate the success of therapy [4]. The Scoliosis Research Society suggests that diagnosis is confirmed when the Cobb angle is greater than 10˚ or the axial rotation is identified [2] using the Perdriolle method [5] or the Nash and Moe method [6].

Treatment options include conservative management (bracing) or operative intervention [2]. In approximately 20% of cases, scoliosis is secondary to another pathological process, while the remaining 80% are cases of AIS [2]. Several studies on the onset and progression of AIS report a higher prevalence and severity of the curve in girls [1–3, 7]. Research is needed to identify gender-specific risk factors [7]. The cause and pathophysiology of AIS appear complex and remain unclear. To date, the causes of scoliosis have mainly been sought in congenital or acquired disorders of the vertebral structure [2]. The causes of scoliosis are supposed to be systemic disorders of, among others, mucopolysaccharide and lipoprotein synthesis [2]. On the one hand, a dysfunction of postural mechanisms involving the central nervous system (CNS) and a distorted body schema have been proposed to be part of a sequence of pathological events in the development of spinal deformity in AIS [8, 9]. On the other hand, CNS alterations have also been explored as a possible cause of AIS [3, 10–15].

CNS involvement can be studied using non-invasive brain imaging techniques such as magnetic resonance imaging (MRI), functional MRI (fMRI) and electroencephalography (EEG). MRI techniques produce detailed three-dimensional anatomical images of the brain. fMRI techniques measure small changes in blood flow that occur with brain activity, whereas EEG measures electrical brain activity. In previous studies, MRI has detected asymmetries in brainstem corticospinal bundles and significant mean volumetric differences in 22 brain regions in people with AIS. These results have led authors to suggest that anatomical alterations of the CNS may be linked to AIS [10, 11]. fMRI studies support the possibility of an underlying neurological disorder in AIS. Abnormal patterns were detected in the motor network during the execution of a simple motor task which consisted in opening and closing one's fist [12]. A decrease in functional connectivity of cortical and subcortical motor structures was found by analyzing AIS resting-state fMRIs, acquired both during extension and semi-flexion of the lower limbs [13]. Not only MRI and fMRI studies, but also EEG studies have revealed abnormal brain activity in AIS. An increase in the amplitude of the peak in the alpha rhythm ([7.5–12.5] Hz) was found in the central, frontal, parietal, and occipital regions when comparing brain activity of adolescents with AIS with that of healthy controls during upright standing

posture [14]. Differences in electrocortical dynamics were found between girls with AIS and controls performing a balance task that involved alteration of ankle proprioception. Specifically, a significant suppression in the power of the alpha ([8–12] Hz) and beta ([13–30] Hz) bands emerged in the AIS group during the ankle vibration interval with eyes open and a significant increase in theta ([4–7] Hz) band power with eyes closed [15]. Alpha oscillations are a marker of the excitability of the somatosensory cortex. Suppression of alpha, compared to baseline, indicates somatosensory cortical activation. Theta rhythm, also involved in balance control, intensifies when control demands increase. EEG data acquired in four different standing positions in adolescent girls with AIS revealed a significant increase in theta activity and lateralized alpha activity during all balance tasks [3].

Parallel to the study of CNS response through non-invasive brain imaging techniques in AIS, human motion analysis provides insight on the quality of balance and movement. Marker-based stereophotogrammetry, inertial measurement sensors, surface electromyography (sEMG) sensors, and force plates, among other systems, can provide valuable information on joint kinematics, muscle activation, center of mass (COM), and center of pressure (COP) displacement. Balance control measured through force plates revealed no differences between girls with AIS and CTRL in a standing upright position [3, 14, 16, 17]. Differences between girls with AIS and CTRL have been detected when the experimenters asked the participants to stand upright while either raising their arms with eyes closed or during sensory cues alteration (i.e., ankle tendon vibration or galvanic vestibular stimulation) [3, 14, 16].

Quantifying EEG, sEMG and motion patterns before, during and after bracing and physiotherapy treatment could be a first step to clarify what triggers the lateral spinal curvature that produces asymmetric loading of the skeletally immature spine, which, in turn, causes asymmetric growth and a progressive wedging deformity [18].

Last but not least, body representation alterations, i.e., alterations in both the *body schema* and the *body image*, need to be better understood in this clinical population. *Body schema* refers to the position and configuration of the human body as a 3-dimensional object in space. The body schema is the sensorimotor representation of the body used to plan and execute movements [9, 19–21]. Alterations of body schema have been proposed to parallel the development of spinal deformity in AIS [8]. *Body image* is a complex psychological construct that encompasses different dimensions, namely perception, cognition, affect, and behavior [22]. Body image refers to the disfiguring appearance caused by scoliosis spinal deformities, and the related brace treatments. Altered body image has a notable impact on the mental health of adolescents [23], which has detrimental effects both on quality of life and adherence to treatment [24, 25].

Body schema is generally assessed using self-reported assessment scales [26]. We propose two ad-hoc tasks to monitor it in a longitudinal study. (i) The *forearm bisection task* is a simple and common test widely used in clinical and research fields to investigate the body schema [27]. It requires the subjects to point to the perceived midpoint of their forearm. Since AIS significantly changes the structure of the adolescent spine, thorax, and trunk, we include an innovative task that involves the movement of the trunk. (ii) The *target reaching* forces the subjects to perform forward trunk movements. It aims to investigate the body schema in adolescent girls with AIS and to determine if alterations in the body schema can modify the planning of movement within the action space immediately surrounding the body, also known as the peripersonal space. The outcomes of the target reaching will be compared with those resulting from the standardized forearm bisection task, to ensure its reliability as a task to evaluate the body schema in adolescents with AIS. During these two tasks, the performance of the subjects will be quantitatively assessed using EEG, marker-based stereophotogrammetry, IMUs, sEMG, and force plates. The EEG signal allows us to quantitatively record the activation of the

sensorimotor cortex where the body schema is 'dynamically' represented. In fact, the representation of the body schema is plastic and can vary [27]. Marker-based stereophotogrammetry will allow us to accurately estimate the timing of each movement and to relate IMU, sEMG and force plate signals with the EEG.

Body image has been largely neglected in clinical practice and is generally assessed with self-reported questionnaires, which are not capable of detecting the complexity of body image alterations. We propose (i) a self-reported questionnaire and (ii) a computer-based test to investigate body image alterations.

Taking into account the results of the state-of-art literature, the primary objective of this experimental protocol is to quantitatively investigate alterations of body representation in AIS. We believe that a quantitative evaluation of body representation alterations in AIS in a longitudinal study may provide new information to better understand the pathophysiology and etiology of scoliosis. The novelty of our study is also based on including the evaluation of girls with AIS before any treatment. We cannot exclude a priori that altered brain activations of the sensorimotor network, altered body schema and image, increased attention toward shoulders and back could be a consequence of treatment per se. Physical therapy exercises and brace treatment could increase proprioception and awareness of the shoulder and back muscles. Our longitudinal study aims to rule out the hypothesis described above.

Results from our previous study, conducted on subjects with AIS who underwent brace treatment, [3] revealed: (i) a significant increase in relative power of EEG delta ([1–4] Hz) and theta rhythms, compared to controls, located in central areas; a significant increase in the laterality index in alpha range observed in adolescents with AIS with main right curve, demonstrating an imbalance between the two sensorimotor areas; (ii) no differences in balance performances between groups; (iii) no differences between controls and adolescents with AIS for the single body segment. However, the investigation of body schema revealed possible alterations in the overall representation of the trunk, namely in the perception of the shoulders-waist proportion, and in its inclination. Based on the literature background and our previous work [3], we expect that, compared to controls, adolescents with AIS, before any treatment, will show: (i) altered brain activation of the sensorimotor network; (ii) greater sway, unbalanced activation of muscles on the right and left sides of the back, and more variable IMU signals on the frontal plane when performing the target reaching, as a result of less efficient control of their balance on the plane with the larger scoliotic deformity; (iii) altered body schema reflecting altered activation of the sensorimotor network.

## Materials and methods

### Experimental design

Adolescent girls with a confirmed diagnosis of mild (Cobb angle ranging from 10˚ to 20˚, no spine surgery) and moderate (Cobb angle ranging from 21˚ to 35˚, no spine surgery) scoliosis (AIS) will be enrolled at the Adolescence Spine Diseases Diagnostic and Therapeutic Center of the Padova University Hospital. Subjects matched in sex and age without AIS—controls (CTRL)—will be recruited through flyers posted in middle and high schools in Padova.

Adolescent girls with AIS and controls will undergo a three-phase experimental protocol (T0, T1, T2).

This longitudinal study aims to follow and quantify alteration in body representation in parallel to the progression of scoliosis from the time of the first diagnosis by the physiatrist (T0) and up to one year after diagnosis (T1, T2). At T0, subjects (AIS and CTRL) will undergo two tasks aimed at investigating possible alterations in *body schema* representation. During the experiment, we will use EEG to record changes in brain oscillations; sEMG to assess possible

imbalances in muscle activation, IMUs, motion capture, and force platforms to monitor the performance and movement characteristics during the balance and motor tasks. A self-reported questionnaire and a computer-based test will be administered to all subjects to investigate *body image* alterations in AIS.

After this evaluation (T0), if the adolescent shows low-grade scoliosis, an expert physiotherapist gives her corrective postural exercises to perform daily (30 min / day) until growth is complete. Participants' performance in scoliosis-specific exercise will be monitored by the physiotherapist at least once a week. The set of exercises is believed to improve postural awareness, proprioception, and ability to dissociate the movements of the lumbar spine from the thoracic one in the different planes of space. Treatment will be planned in the individual patient, based on the dysfunction detected in the clinical assessment. When scoliosis exceeds the 20°Cobb and there is evidence of progression risk, a rigid partial (18 hours/day) or full-time brace (23 hours/day) is also prescribed.

The same physical, neurological, and cognitive evaluations of T0 will be repeated after 6 (T1) and 12 months (T2) of therapy in AIS and after 12 months (T2) in CTRL. A team of bioengineers, physiatrists and physiotherapists will perform the experimental protocol to ensure the comfort of the subjects and the quality of the data (Fig 1).

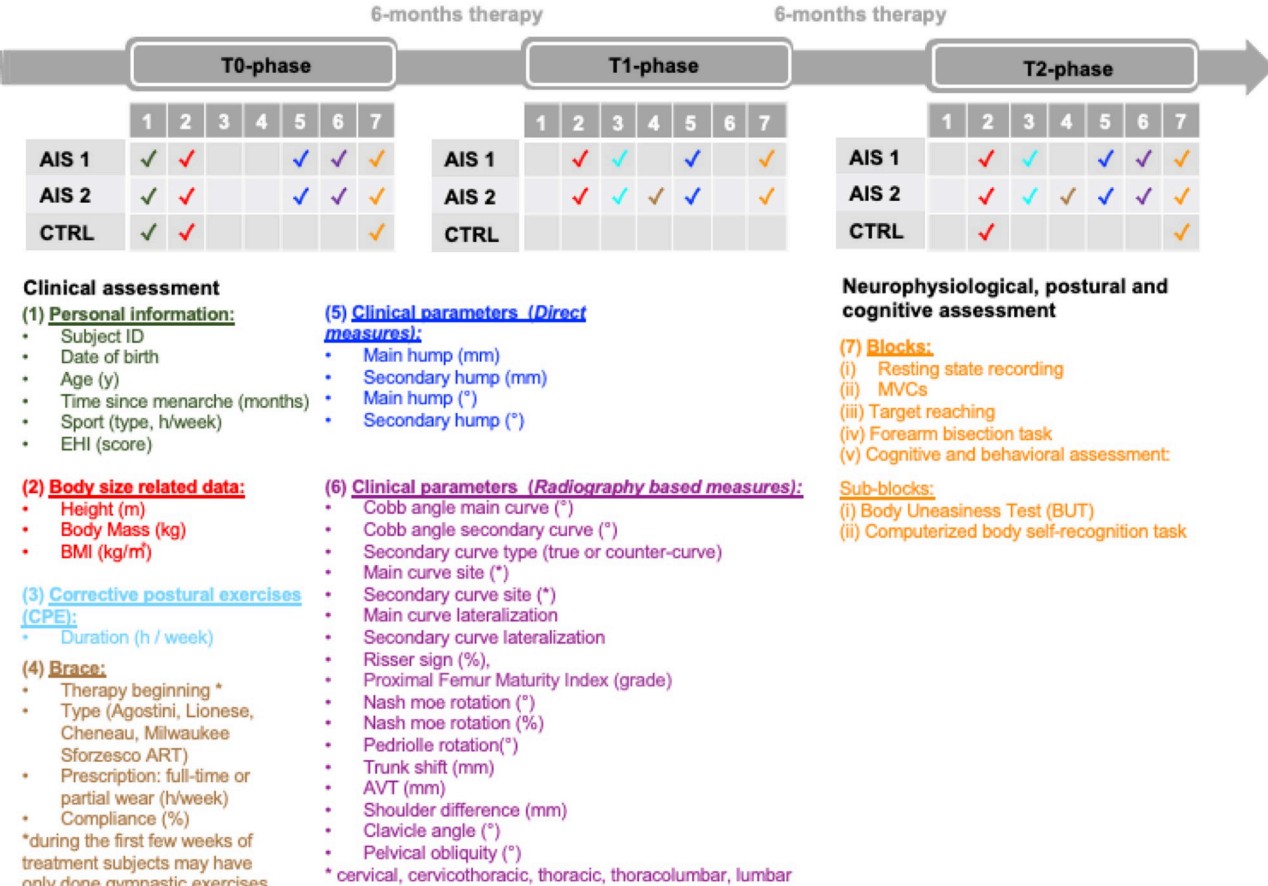

**Fig 1. Timeline.** Clinical, neurophysiological, postural and cognitive assessment timeline for adolescents girls with AIS assisted with corrective postural exercises (AIS 1), adolescent girls with AIS treated with both corrective postural exercises and partial or full-time brace (AIS 2) and CTRL. Tables report for each group and for each phase (T0, T1, T2) which parameters will be recorded. The personal and clinical parameters and experimental protocol block descriptions are reported in the lower panel (1), (2), (3), (4), (5), (6), (7).

**Materials and equipment.** Motion capture data will be acquired from our 10-camera stereophotogrammetric system at 100 Hz. The motion capture system includes a control box (Lock Synch Box, Vicon) to synchronously record analog force plate data (Bertec 4060–10, Bertec Corporation, Columbus, Ohio, United States), IMU and sEMG data (WaveTrack IMU and MiniWave, Cometa srl, Milan, Italy), which will be sampled at 2000 Hz. The sEMG signals will be filtered between 20–450 Hz. EEG data will be acquired from 64 channels at a sampling frequency of 500 Hz and will be referenced to Cpz (ANT Neuro, Enschede, The Netherlands). The synchronization between the stereophotogrammetric system and the portable EEG system will be guaranteed by a custom-made MATLAB script, which instructs and uses a trigger box (BrainTrends, Rome, Italy) to send: (i) the start acquisition command to the Vicon system (responsible for stereophotogrammetric, EMG, IMU and force platforms data collection); and (ii) a time stamp to the EEG system, which is already recording.

## Procedure

All methods will be carried out in accordance with the guidelines of the 2008 Helsinki Declaration. Ethical approval has been obtained on April 2023. Written informed consent will be provided by both subjects and their legal guardians.

We estimate recruiting 40 subjects. Considering a 20% dropout (approximately 8 subjects), we expect to acquire data from 30 subjects (15 subjects with AIS and 15 CTRL). For the calculation, we used the average effect size observed in [3] (i.e., the minimum difference between CTRL and AIS subjects for the EEG laterality index in sensorimotor areas) considering error probability ($\alpha$) of 5%, with a power (1 –$\beta$ error probability) of 80%. Power statistical analyses will be repeated after having recorded data at T0 as an additional tool to review the required sample size. If necessary, the sample size will be enlarged.

**Selection criteria.** Control subjects—Inclusion criteria:

- sample size: N $\geq$ 15 subjects

- sex: female

- age range: 11–16 years

- right-handedness

- normal body mass index (BMI) (between 18.5 and 24.9 kg/m$^2$)

  Control subjects—Exclusion criteria:

- spinal pathology or any known neurological or musculoskeletal disorders

- clinically relevant back hump

- presence of any trunk alteration

- intake of any drug with a central nervous disease effect

- dermatologic problems

- sporting activity at competitive level ($\geq$ 10 h/week) or involving balance-enhancing activities

  AIS subjects—Inclusion criteria:

- sample size: N $\geq$ 15 subjects

- sex: female

- age range: 11–16 years

- right-handedness

- with mild (Cobb angle ranging from 10˚ to 20˚, no spine surgery) and moderate (21–35˚, no spine surgery) scoliosis

- normal BMI (between 18.5 and 24.9 kg/m$^2$)

    AIS subjects—Exclusion criteria:

- severe scoliosis (Cobb angle > 35˚, possible spine surgery required)

- already treated with brace or corrective postural exercises

- intake of any drug with a central nervous disease effect

- dermatologic problems

- sporting activity at competitive level ($\geq$ 10 h/week) or involving balance-enhancing activities

**Clinical assessment.** Before undergoing the experimental protocol, the following data will be recorded for each subject: age, weight, height, BMI, sexual maturity, evaluated according to information related to menarche, and physical activity. Handedness will be evaluated using the Oldfield Questionnaire [28].

During their first medical examination, adolescents with AIS will undergo a physical assessment performed by a physiatrist to (i) detect the presence of possible body asymmetries and (ii) measure the paravertebral humps in the forward bent position with the Bunnell scoliometer and the Ferraro humpmeter [29, 30]. All subjects will be evaluated by the same physiatrist to ensure data repeatability and minimize intersubject measurement error. A standing full-spine posterior-anterior x-ray will be performed. The curve entity will be quantified using the Cobb angle method [31], while the Perdriolle method will be applied to estimate the rotation of the apical vertebrae [5]. Based on the radiological image, the physiatrist will also collect other relevant clinical measures, i.e., the initial correction rate (ICR), the side and anatomical site of the convexity of the scoliosis, trunk asymmetries, shoulders, and hips inclination. Radiological examination will be repeated only at T2 to avoid excessive radiation exposure in adolescent girls, allowing clinicians to quantify possible variations in the values of the parameters. The progression of AIS will be defined as a worsening of more than 5˚Cobb in the magnitude of the curve during the duration of the study (approximately one year). Therapy specifics (e.g., brace type, prescription, compliance, duration of exercise) will be reported both for subjects who will be treated with a partial or full-time brace and for those who will only be assisted with corrective postural exercises (for more details, see Fig 1).

**Experimental protocol.** Subjects will be asked to wear a top stretch and a pair of shorts. Before starting the recordings, snap-lead sEMG sensors will be placed bilaterally on the belly of the muscle of the erector spinae, latissimus dorsi, and abdominal muscles (rectus abdominis, abdominal external oblique, abdominal internal oblique) as indicated in S1 Appendix. Sensors will be placed along the direction of the muscle fiber after shaving the skin to remove excess hair, and cleaning the skin with an alcohol swab to remove sweat and dirt. The quality of the sEMG signal will be ensured by visually inspecting the signals for low baseline noise at rest and the absence of low-frequency motion artifact during movement [32]. Wearable IMU devices (accelerometer, magnetometer, and gyroscope) will be placed with a double-sided adhesive tape on the fifth lumbar vertebra (L5) and bilaterally on the tibia. The impedance values of the

EEG sensors will be recorded, checked, and stored at the beginning and end of the experiment. The whole protocol will last approximately 80 min and will be repeated once at each experimental phase. The protocol consists of five blocks. First, subjects will undergo (i) a resting state recording as a baseline and (ii) Maximal Voluntary contractions. Second, the body schema will be investigated through tasks based on the (iii) target reaching, and (iv) forearm bisection task. Blocks (iii)-(iv) will be administered in random order. Eventually, the perception of body image by subjects will be investigated through (v) a cognitive and behavioral evaluation (Fig 1). We refer the reader to S1 Appendix for a detailed description of the implementation of tasks.

*(i) Resting-state recording (6 min).* will be performed before executing the two tasks as a baseline condition. A 6-min EEG resting state recording of quiet upright standing (without shoes) will be acquired in a sound-attenuated room, first with eyes open, looking at a fixation cross ($\sim$ 3 m in front of the subjects), and then with eyes closed. Participants will be instructed to keep their feet position parallel and at shoulder width. COP sway will be recorded using a force platform. Acceleration and angular velocity data gathered from the IMU placed on the lower back will be processed to estimate pelvis motion during the standing trial and quantify participants' balance control performance [33, 34]. Pelvis kinematics will also be evaluated through the motion capture system. This will allow checking whether the use of a waist-mounted IMU produces reliable quantities to assess AIS subjects stability in clinical settings, without demanding for a motion capture laboratory.

*(ii) Maximal Voluntary Contractions (MVCs) (10 min).* will be performed to estimate the maximum sEMG amplitude for signal normalization. Subjects will be asked to perform isometric contractions against resistance as detailed in S1 Appendix to maximally activate each muscle tested. 5 min of rest between trials will be provided to avoid fatigue accumulation. We will also ask the subject if she may need longer rest time. Subjects will be carefully instructed, trained, and encouraged to ensure maximum exertion. (The placement of electrodes on the muscle and the procedures for acquiring a voluntary movement contraction (MVC) of each muscle are detailed in S1 Appendix).

*(iii) Target reaching (15 min).* The task will assess the influence of body schema on the planning of the movement. Subjects will start with their hand on their chest and observe the experimenter, standing next to them, touching (for 3 s) one of the 10 targets (represented in Fig 2A) displayed with different shapes and color and placed at different x-y position on the board. Then, they will be instructed to reach the same point before returning to the starting position with the hand at the chest, keeping their eyes closed. The 10 targets will be presented to the subjects in a randomized order. The same task will be repeated both with the dominant and with the non-dominant hand. The Euclidean distance between the x-y coordinates of the two points (experimenter- and subject-selected) will be calculated for each target as a measure of the precision of task execution (see S1 Appendix for further details).

*(iv) Forearm bisection task (10 min).* The task will be used to test the representation of the metric properties of the body. In each trial, subjects will be repeatedly asked to point, with the index finger of the non-dominant hand, the midpoint of their dominant forearm (Fig 2B). Subjects will be required to return the non-dominant arm to the starting position at the end of each trial and to wait for a start signal from the experimenter to perform the movement. The task will be repeated 10 times. The midpoint of the subjects' arm will be calculated as the difference between the y-coordinates of the markers placed on the dominant arm. The distance between the midpoint of the subject's arm and the selected location of the subject will be calculated as a measure of movement accuracy (see S1 Appendix for details).

*(v.i) Cognitive and behavioral assessment—Body Uneasiness Test (BUT—10 min).* To test body image self-attitude, we will administer the Italian validated version of BUT [35]. BUT is a

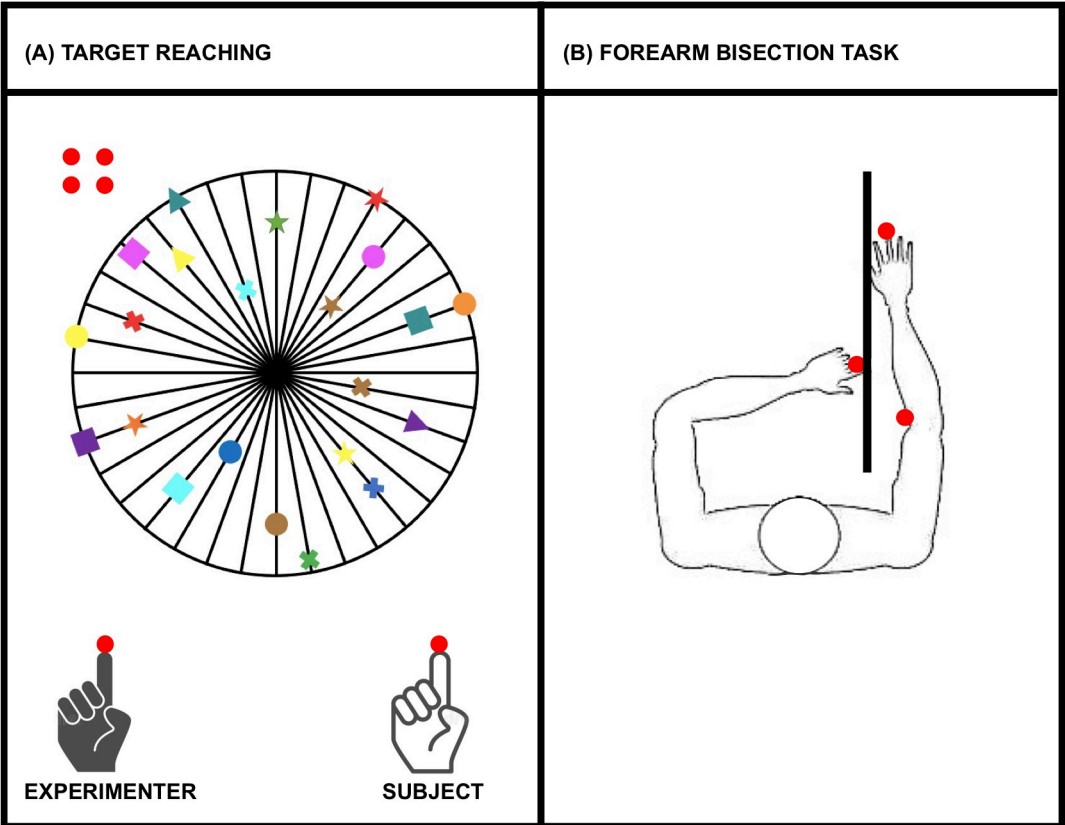

**Fig 2. Blocks (iii)-(iv) graphical representation.** (A) target reaching; (B) forearm bisection task. The red circles symbolize the passive retroreflective markers. The subjects' hand is represented in white and the experimenter's hand is represented in black.

71-item self-report questionnaire that includes a first part (BUT-A) assessing body image concerns, weight phobia, avoidance behaviors, compulsive self-monitoring, detachment, and estrangement feelings toward one's own body (34 statements), and a second part (BUT-B) which looks at worries related to specific body parts. Each item is rated on a six-point scale ranging from 0 (never) to 5 (always).

*(v.ii) Cognitive and behavioral assessment—Computerized body self-recognition task (15–20 min).* Selective attention to body parts that do not like (attentional bias) will be assessed using a computerized body self-recognition task (readaptation of the task adopted by [36]). During the task, subjects, seated in front of a monitor, at a distance of about 30 cm, will be presented with three vertically aligned square grayscale images depicting the body parts of other people or their own body parts. The body parts presented could be disease relevant (i.e., shoulders, back) or irrelevant (i.e., leg, arm, hand, foot). Subjects will be instructed to decide as accurately and fast as possible which of the upper or lower images is of the same person as the central target stimulus presented in a red frame. The duration of the trial will not be limited and no pressure will be exerted on the performance of the subjects. Accuracy and reaction times will be collected as outcome variables. See S1 Appendix for more details.

**Foreseen data processing.** We will investigate both linear (i.e., power spectral density [37], time-varying connectivity [38]) and nonlinear EEG features (i.e., entropy and fractal properties [39, 40] in the scalp and inverse domains (Electrical Source Imaging [41]) from the

data recorded at rest and during the execution of target reaching. The laterality index, which describes the contrast in amount of activation (i.e., relative power in the alpha band) between the right and left hemispheres, will be calculated during the target reaching.

In parallel to the quantification of EEG characteristics, the Root Mean Square (RMS) value will be estimated for each sEMG signal, normalized by the MVC, to compare muscular activity within and between groups and investigate possible imbalances in activation patterns between the right and left sides of the body. sEMG signals acquired from the trunk and abdominal muscles during the resting state and the target reaching tasks will allow us to quantify a possible imbalance in the muscle activation pattern between the left and right sides of the body, as reported in previous studies.

Sensors in the trunk and legs will be used to assess the sway and oscillations of subjects at the level of the upper and lower body, respectively, during resting state and the target reaching. Force plates will measure the displacement of the COP, and the motion capture system will allow tracking of arm movement during the target reaching and forearm bisection tasks. In particular, COP data will be used to measure the range of motion in the anterior-posterior (AP) and medial-lateral (ML) directions, the Sway Path length (SP) and the area of the ellipse containing the 95% of the COP trajectory to assess balance during the resting state and target reaching tasks [42, 43]. The data gathered from the waist-mounted IMU [39, 44–46], will be used to evaluate an index of performance quality, complexity and smoothness of motion. To assess possible body schema alterations we will evaluate the task execution accuracy, calculating the Euclidean distance between the actual and the subject selected targets (see S1 Appendix for details).

To reveal possible alterations in body image representation in adolescent girls with AIS, BUT sub-scores will be calculated (scoring instructions are reported in S1 Appendix). Response accuracy (i.e., percentage number of correct responses over the total number of trials) and reaction times (RTs), collected as outcome variables during the Computerized body self-recognition task, trials will be clustered according to the target stimulus and subsequently analysed. The correlations between the BUT questionnaire subscores and RTs and accuracy will be assessed using the Pearson's correlation coefficient (r).

The results of AIS-adolescent girls and controls of the same age and sex will be compared by statistical analyses. Shapiro-Wilk normality test will be applied on all the data collected to check the normal distribution. A two-tailed paired sample t-test or a two-sided Wilcoxon rank sum test ($p < 0.05$) will identify differences between the two groups (AIS vs. CTRL) and between the three time points evaluation (T1, T0 and T2) after having run a global test (either repeated measure ANOVA, or Friedman test depending on data normality).

**Protocol feasibility assessment.** At the end of the experimental session, a survey will be administered to assess the complexity, length, and feasibility of the entire protocol. If the experience is considered too frustrating and tiresome, a simplification or reduction of the two tasks can be adopted after the first pilot recordings to increase both the comfort and interest of the young subjects.

**Management.** Participation to the study is voluntary and is always allowed to withdraw at any time without consequences. Each subject and their legal representatives will receive a privacy information sheet and a consent form. Subjects will be volunteers by self-selection and have sufficient cognitive status to provide informed consent, so vulnerable groups will not be considered in this research program. All subjects will have a one week washout period to consider participation in the study before informed consent is obtained. Once engaged in this longitudinal study, subjects will undergo the first physiatric visit. Confidentiality will be maintained throughout this study. All data collected by the researchers will be anonymous or anonymised prior to analysis. Demographic data on the participant will be included in the

analysis of data, but this be presented in a way that provides no means to identify a natural living person.

There are no risks of harm to disclose, since all the measures are non-invasive.

The confidentiality and anonymity of the data will be guaranteed. At the end of the research period, the research data will be retained for further analysis and will not be sent to any third party.

All the staff involved in the research will be specialized personnel (e.g., physiatrists, neuropsychologists, physiotherapists, bioengineers), trained for the purposes of the research.

The ethical issues associated with the participation of (human) volunteers will be addressed through full compliance with the protection of personal data regulations, unobtrusive procedures, and maximum care for safety issues during field trials according to regulations in Italy and Europe.

**Project timeline.**   Since this experimental protocol has already been approved from the Ethics Committee on April 2023 (n. 5627/AO/22), the study started in June 2023 and lasts two years. The laboratory setup will require two months (months 1–2). Data acquisition will be performed before treatment (T0) with the full-time brace (T0—months 3–7) and after 6 and 12 months (T1—months 9–13 and T2—months 15–19) full-time brace treatment and scoliosis-specific exercises (i.e., rehabilitation intervention—months 3–19). In parallel to subject recruitment (months 1–5) and data acquisition (months 3–7, 9–13, and 15–19), signal processing and interpretation (months 4–22) will be carried out to lead to the final stages of the project (months 20–24). The dissemination of results will not be limited to the final months of the project, but preliminary results after T0-data recording and processing (months 3–7) will be shared with academic and non-academic audiences through conference proceedings, seminars, and journal articles within the first year (months 9–13).

## Discussion

The innovative aspect of the proposed experimental protocol is the quantitative evaluation of girls with AIS before any treatment based on the combination of neurophysiological, postural and cognitive / behavior analyses, during the execution of exercises related to the body schema adapted for subjects with AIS. These will help us (i) investigate the neurophysiological processes underlying AIS and highlight neurophysiological biomarkers of disease progression; (ii) evaluate balance and target reaching performance and (iii) determine a routine battery of tests to monitor body representation alterations (i.e., body image and body schema) before treatment (i.e., partial or full-time brace and corrective postural exercises) and during/after treatment.

In our preliminary study [3], an increased theta activity and lateralized alpha activity were observed in central areas in AIS-adolescent girls during balance tasks. Increased theta power within central clusters may reflect a higher information processing load due to the increased postural demands caused by scoliosis, and alpha lateralization might be interpreted as a compensatory strategy to overcome sensorimotor dysfunction mirrored by altered body schema [3, 47, 48].

At T0, we expect significant differences in the EEG rhythms associated with postural control between adolescents with and without AIS, during upright stance and target reaching.

Our longitudinal protocol may highlight EEG alterations in AIS after 6 and 12 months of targeted therapy. We expect an improvement in the clinical condition of the subjects, produced by therapy, associated with neurophysiological changes: a decrease in theta rhythm due to decreased postural demands, and a reorganization of alpha rhythm over the sensorimotor areas.

Balance during upright stance is maintained through an accurate sensory input fusion integrated at different levels of the CNS and producing appropriate motor output. Postural control mechanisms in subjects with AIS need to be investigated further. Our previous study did not show balance alterations in adolescent girls with AIS, compared to healthy controls, when standing in upright position [3]. In the proposed experimental protocol, we plan to conduct a more in-depth analysis, evaluating balance conditions in subjects with AIS and CTRL, using data from IMU and force platforms. On the one hand, we expect similar balance performances in AIS and CTRL during the resting-state, complemented by unbalanced activation of the muscles on the right and left sides of the back, which may validate our hypothesis that adolescent girls with AIS adapt their brain activity to prevent scoliosis-causing large body movement. On the other hand, we foresee that the target reaching will highlight abnormalities in tuning the postural control in AIS compared to CTRL. Such abnormalities, combined with the mechanical effect of the scoliotic deformity, are expected to result in altered muscle activation symmetry, greater instability, and need for postural adjustments in AIS compared to CTRL groups. These findings will pave the way for the discovery of alterations in postural control mechanisms in adolescent girls with AIS.

Body schema refers to the sensorimotor body representation that entails tracking and updating the position and configuration of both body parts and other objects in space towards which reaching movements can be planned [27]. The functional representation of the action space immediately surrounding the body, also known as the peripersonal space, usually undergoes changes concurrent with body schema alterations. Through the target reaching we will assess if possible body schema alterations in adolescent girls with AIS can alter the planning of the movement within the peripersonal space. The forearm bisection task is a simple and common test widely used in clinical and research fields to investigate the body schema [27]. We will use this task to test whether adolescent girls with AIS have a different representation of the metric properties of their own body compared to CTRL. By replicating the same tasks at time T1 and T2, we will test how the body schema will change as a function of control postural exercises or the effectiveness of brace therapy. The first test of the body schema (T0) is expected to highlight a possible alteration in the perception of the shoulders-waist proportion and in the inclination of the trunk. We expect a decrease in alteration in their body representation due to the treatment. The two tasks are designed to reveal body schema alterations at T0 in the AIS group, which should improve significantly after the 6 month exercise and brace-treatment period (T1).

Cognitive and behavioral assessment will investigate two aspects of body image perception: how subjects perceive their body appearance and the emotional and cognitive aspects of body [26]. We expect to find significant differences between the AIS group compared to CTRL at T0, with AIS having a worse self-attitude towards body image (i.e. body image dissatisfaction). Worst score in the sub-domains of global severity index, body image concerns, avoidance and compulsive self-behaviors (BUT-A), and specific higher concerns for shoulders, chest, and hips than other body parts (BUT-B). Second, we expect to find body attentional bias in the AIS group towards disease-specific body parts (shoulders, back), meaning faster reaction time and higher accuracy in response to disease-relevant images, and particularly to own body-relevant images. At T1, we expect to find a significant improvement in self-attitude scores for body image compared to T0 and a reduction of the attentional bias towards disease-relevant body parts, which means no differences in reaction time and accuracy between relevant and irrelevant body parts.

A risk of the proposed experimental protocol is boredom and loss of interest and compliance due to its duration (circa 80 min) and complexity. If necessary, the protocol can be divided into three separate sessions on three different days. Block (i), blocks (iii-iv), and block

(v) of the protocol are independent and investigate different research questions. Another possible risk is related to the longitudinal design. Adolescent girls with AIS may not be available to repeat neurophysiological, postural, and cognitive assessment after 6 and 12 months. To mitigate this risk, we will make our best efforts to retain subjects through periodic follow up; and we will increase the number of recruited individuals at T0 to take into account an approximately 20% dropout rate. Although not all individuals may complete all follow-up sessions, the protocol will still provide valuable data that will allow us to investigate the muscular and neurophysiological correlates of AIS compared to CTRL at enrollment (T0). The forearm bisection task, block (iv), and resting-state EEG recording can still provide us significant information regarding possible body schema alterations, neurophysiological patterns, and postural performances.

## Conclusion

Investigating a defective postural-CNS control and body representation alterations in AIS is a key step toward the identification of the pathogenic mechanism underlying scoliosis. The proposed protocol has utility for future studies in which datasets of different nature should be collected and interpreted in adolescent girls with AIS. This experimental protocol that aims to integrate cerebral, muscular, postural, cognitive, and behavioral data may (i) enrich current routine exams performed in clinical practice, (ii) offer novel therapeutic targets according to identified pathophysiological patterns, and (iii) contribute to advance scientific knowledge in neuroscience, musculoskeletal disorders, and rehabilitation.

## Supporting information

**S1 Appendix. Guidelines to reproduce the experimental protocol.**
(PDF)

## Author Contributions

**Conceptualization:** Matilde Paramento, Maria Rubega, Roberto Di Marco, Paola Contessa, Michela Agostini, Francesca Cantele, Stefano Masiero, Emanuela Formaggio.

**Funding acquisition:** Stefano Masiero, Emanuela Formaggio.

**Investigation:** Matilde Paramento, Maria Rubega, Paola Contessa, Michela Agostini, Emanuela Formaggio.

**Methodology:** Maria Rubega, Roberto Di Marco, Emanuela Formaggio.

**Project administration:** Emanuela Formaggio.

**Resources:** Stefano Masiero.

**Supervision:** Stefano Masiero, Emanuela Formaggio.

**Writing – original draft:** Matilde Paramento, Maria Rubega, Emanuela Formaggio.

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
