## [Decision Letter · Decision Letter 0]

10 Jul 2023

PONE-D-23-08292

Experimental protocol to investigate cortical, muscular and body representation alterations in adolescents with idiopathic scoliosis

PLOS ONE

Dear Dr. Rubega,

Thank you for submitting your manuscript to PLOS ONE. After careful consideration, we feel that it has merit but does not fully meet PLOS ONE’s publication criteria as it currently stands. Therefore, we invite you to submit a revised version of the manuscript that addresses the points raised during the review process.

ACADEMIC EDITOR:

Authors have presented a good quality paper, however, there are still points that have to be clarified and require a major revision of the manuscript. In particular, Authros have to clarify the rationale bheind the aim of the study and of the methodologies employed as also suggested by the reviewers.

We look forward to receiving your revised manuscript.

Kind regards,

Andrea Tigrini, Ph.D.

Academic Editor

PLOS ONE

Journal Requirements: 

4. We note that you have stated that you will provide repository information for your data at acceptance. Should your manuscript be accepted for publication, we will hold it until you provide the relevant accession numbers or DOIs necessary to access your data. If you wish to make changes to your Data Availability statement, please describe these changes in your cover letter and we will update your Data Availability statement to reflect the information you provide

Additional Editor Comments :

Authors have presented a good quality paper, however, there are still points that have to be clarified and require a major revision of the manuscript. In particular, Authros have to clarify the rationale bheind the aim of the study and of the methodologies employed as also suggested by the reviewers.

Reviewers' comments:

Reviewer's Responses to Questions

**Comments to the Author**

1. Does the manuscript provide a valid rationale for the proposed study, with clearly identified and justified research questions?

Reviewer #1: Yes

Reviewer #2: Partly

2. Is the protocol technically sound and planned in a manner that will lead to a meaningful outcome and allow testing the stated hypotheses?

Reviewer #1: Partly

Reviewer #2: Partly

3. Is the methodology feasible and described in sufficient detail to allow the work to be replicable?

Reviewer #1: Yes

Reviewer #2: Yes

4. Have the authors described where all data underlying the findings will be made available when the study is complete?

Reviewer #1: Yes

Reviewer #2: No

5. Is the manuscript presented in an intelligible fashion and written in standard English?

Reviewer #1: No

Reviewer #2: Yes

6. Review Comments to the Author

You may also provide optional suggestions and comments to authors that they might find helpful in planning their study.

Reviewer #1: I have the following comments regarding the protocol:

1) Sample size appears relatively small, especially if future statistical analyses requiring further subgrouping. I understand that the sample size proposed was based on the difference between control and AIS detected in another study as cited. What if the difference detected is smaller than what was found in that study? Please provide plans of power analyses and whether there is a need of review of sample size as an interim measure for this study.

2) Sexual maturity and Risser sign were collected as maturity measures - Risser sign has been known for its insensitivity and inaccuracy of its use in AIS. I would suggest the authors to consider additional skeletal maturity measure that will be available for assessment on whole spine radiographs such as the Proximal Femur Maturity Index (PFMI).

3) The duration of 80 minutes of tests appear quite long for adolescents, plus the additional Protocol feasibility assessment at the end of the experimental session - please consider adopting the separate sessions approach as proposed, or to condense the experimental session.

4) Regarding brace-wear compliance - is there an objective measure of compliance such as by thermal sensor in the brace? Brace-wear compliance can have an impact on the muscles and thus can be affecting the study results.

Also wondering if there is any measure to take into account brace correction rate and correct brace-wear at how many hours in a day (%).

5) There has always been an issue regarding how accurate patients can perform scoliosis-specific exercise exactly on their own and the accuracy of exercise duration reports. Again this aspect of the exercise can potentially affect study results. Any thoughts by the authors on that?

6) This study emphasizes on quantifying EEG and sEMG signals during the progression of AIS - the authors should define how they evaluate and define curve progression in other to achieve the above.

Reviewer #2: This study protocol is interesting as several measures will permit assessing different neural mechanisms across time. However, the authors are not sufficiently clear about what they will analyze, and they do not present the rationale for proposing these tasks.

Here are my comments concerning this study protocol.

P2 – L36: Should mention/explain the task rather than simply state it is a simple task.

P3- L60: I am surprised by this statement as several studies reported balance control impairment in AIS compared to controls during sensory manipulation while participants stood upright. I encourage the authors to scan the literature.

P3 – L93: The authors should explain what the goal of these tasks is. How will the results of these tasks allow the authors to identify the neural mechanisms related to curve progression or can trigger the Stoke's Vicious Cycle Pathogenesis?

P4 – L97: I encourage the authors to be more specific here. In the present form it is unclear whether the authors have specific hypotheses. The dimension for exploring neurophysiological and muscular changes is infinite! How tracking will be realized?

P4 – L104: What treatment the authors are referring to? In addition, will all participants have the same treatment? If not, how will the authors interpret changes across time if the treatment differs among the participants?

P4 – L105: I am confused; will the authors offer participants psychological support or cognitive behavior therapies?

P4 – L108: I agree with these hypotheses, as several previous studies have reported these changes between controls and AIS. However, how will these results bring novel knowledge and allow the authors to identify biomarkers causing curve progression or trigger Stoke's Vicious Cycle? This is unclear!

P4 – L119: This is interesting! Can the imbalance between the two sensorimotor areas be a biomarker of curve progression? I encourage the authors to state whether they have specific hypotheses. In addition, I am not sure to what groups the authors are referring to as at the beginning of this sentence it is stated that the authors studied static balance in participants with AIS only.

P4 – L128: What is the main objective of the functional upper limb tasks. Is it only to induce greater postural challenges? The authors must explain what are the mechanisms that this particular task will permit to study.

P4 – L132: You reported the range of Cobb's angle for mild and moderate scoliosis above. However, you need to report these ranges here also as it is the section describing the participants and the experimental design.

P4 – L140: You need to explain the task permitting to assess body schema in static and dynamic conditions.

P5 – L145: You should report the muscles and the rational for measuring the muscle activities of these muscles. Many previous studies have reported differences in back muscles. Are you going to record muscle activation from different muscles?

P5 – L152: "…during the proposed task…". Explain the task here to help the readers follow you story.

P5 – L154: "… during tasks." Again, explain the tasks here to help the readers follow you story.

P5 – L158: Will these postural exercises be the same for all participants regardless of the type of scoliosis and the severity of the deformation?

P5 – L171: I am puzzled here as the sampling frequency of the IMU is 142 Hz and this is not a multiple of the sampling frequency of the Vicon which is 1000 Hz. Thus, I am not sure that it is possible to resample up the IMU data to 1000 Hz or to down sample the Vicon data to 142 Hz! Can the authors explain how this is possible?

P5 – L183: "Before starting any activity, Ethical clearance will be obtained." This sentence is confusing as below, it is stated that the Ethics committee has approved the research protocol.

P7 – L256: How long will last the resting-state recording? In addition, the authors need to explain how the maximal voluntary contractions will be performed and to measure the MVC of what muscles. Later, there are paragraphs describing the resting-state recording and the measure of the MVC. The authors should consider reorganizing this section as it does not provide valuable information for the resting-state and MVC measures.

P7-L263: Why is recording resting-state EEG with and without visual cues important?

P7- L266: This is unclear as to estimate the whole body center of mass (COM), one must calculate each segment COM's position. Then, the algebraic sum of all individual COM represents the whole-body COM. It is unclear to me how acceleration and angular velocity of the segments can help calculate the whole-body COM. If I am wrong, I encourage the authors to explain, at least to me, how acceleration and angular velocity signals can be used to estimate whole-body COM.

P8 – L272: What is sufficient rest?

P8 -L275: This paragraph should start with a sentence explaining this task's aim and what neural mechanisms this task will permit to study.

P8 – L285: Again, this paragraph should start with a sentence explaining this task's aim and what neural mechanisms this task will permit to study.

P9 – L336: What are these dynamic tasks?

P10 – L364: The authors reported this information above.

P10 – L368: Is this sentence necessary?

P10 – L391: Above L183, I understand that the Ethics Committee will evaluate the experimental protocol! Thus, there is some confusion here!

7. PLOS authors have the option to publish the peer review history of their article (what does this mean?). If published, this will include your full peer review and any attached files.

Reviewer #1: No

Reviewer #2: No

---

## [Author Response · Author response to Decision Letter 0]

5 Aug 2023

Dear Editor,

We would like to thank the Reviewers for the careful evaluation of our work, as well as for the constructive comments and suggestions they provided which improved and enriched our work. We have considered each of the comments and revised our manuscript accordingly.

In the file labeled “Response to Reviewers”, we provide a point-by-point response (text in blue) to each of the comments (text in black). The related changes in the manuscript are in red.

The manuscript has been substantially revised following the suggestions of the Reviewers. Specifically, thank you to the Reviewers’ feedback:

We clarified the rationale behind the aim of the study: The primary aim of this experimental protocol is to quantitatively investigate alterations in the body representation in AIS, and to quantitatively and objectively track the modifications in the sensorimotor representation of the body due to the treatment;

We clarified the rationale behind the methodologies employed: 1) The forearm bisection task is a simple and common test widely used in clinical and research fields to investigate the body schema; 2) The target reaching forces the subjects to perform forward trunk movements to investigate if alterations of body schema can modify the planning of the movement within the action space immediately surrounding the body. The instrumentation of these two tasks will allow us to quantitatively track the changes in body schema.

We hope that the manuscript will be now suitable for publication in PLOS ONE.

On behalf of all authors,

Kind regards,

Maria Rubega, Ph.D.

---

## [Decision Letter · Decision Letter 1]

21 Sep 2023

PONE-D-23-08292R1Experimental protocol to investigate cortical, muscular and body representation alterations in adolescents with idiopathic scoliosisPLOS ONE

Dear Dr. Rubega,

Thank you for submitting your manuscript to PLOS ONE. After careful consideration, we feel that it has merit but does not fully meet PLOS ONE’s publication criteria as it currently stands. Therefore, we invite you to submit a revised version of the manuscript that addresses the points raised during the review process.

ACADEMIC EDITOR: Although the paper was cnsistently updated, Authors should solve minor points to make the paper availabel for publication. Thus Authros have to clarify both the major and minor issues highlighted by the Reviewers.

We look forward to receiving your revised manuscript.

Kind regards,

Andrea Tigrini, Ph.D.

Academic Editor

PLOS ONE

Journal Requirements:

Additional Editor Comments:

Although the paper was cnsistently updated, Authors should solve minor points to make the paper availabel for publication.

Reviewers' comments:

Reviewer's Responses to Questions

**Comments to the Author**

1. Does the manuscript provide a valid rationale for the proposed study, with clearly identified and justified research questions?

Reviewer #2: Yes

2. Is the protocol technically sound and planned in a manner that will lead to a meaningful outcome and allow testing the stated hypotheses?

Reviewer #2: Yes

3. Is the methodology feasible and described in sufficient detail to allow the work to be replicable?

Reviewer #2: Yes

4. Have the authors described where all data underlying the findings will be made available when the study is complete?

Reviewer #2: No

5. Is the manuscript presented in an intelligible fashion and written in standard English?

Reviewer #2: Yes

6. Review Comments to the Author

You may also provide optional suggestions and comments to authors that they might find helpful in planning their study.

Reviewer #2: The authors have improved their manuscript, it is now easier to follow the research protocol and the rational for carrying out this study.

I have a minor suggestion and a major concern that need to be addressed by the authors.

Minor suggestion (L65): “… while engaged in a second task (i.e., applying vibration to a muscle tendon…)…”. Electrical vestibular stimulation or tendon vibration are not a secondary task. I suggest rephrasing: "... while either raising their arms with eyes closed or during sensory cues alteration (i.e., ankle tendon vibration or galvanic vestibular stimulation)."

Major concern: To assess the kinematics of the center of mass (COM), the authors indicated that they will use data from an IMU located at the waist. This sentence is supported by two references. Looking at the paper of Germanotta et al., (2021), I am surprized that the authors propose to use the strapdown integration to estimate the kinematics of the center of mass. In Figure 6 of the paper of Germanotta et al., (2021), the reported errors (RMSE) between the strapdown integration method and the optoelectronic system are ~ 50 mm and ~ 40 mm/s for postural sway and sway velocity along the anteroposterior axis. The amplitude of these errors is larger for postural sway along the mediolateral axis. Since the calculation of the center of mass requires accurate estimate of the trunk center of mass, it is likely that that these errors will be larger when estimating the center of mass of girls with idiopathic scoliosis, as the estimation of the trunk center of mass is difficult to calculate in person with idiopathic scoliosis.

7. PLOS authors have the option to publish the peer review history of their article (what does this mean?). If published, this will include your full peer review and any attached files.

Reviewer #2: No

---

## [Author Response · Author response to Decision Letter 1]

28 Sep 2023

Dear Editor,

We would like to thank the Reviewer for the positive feedback. We have considered both the comments and revised our manuscript accordingly.

In the file labeled “Response to Reviewers”, we provide a point-by-point response (text in blue) to each of the comments (text in black). The related changes in the manuscript are in red.

Specifically, thank you to the Reviewer’s feedback:

We rephrased as suggested the lines 64-67 at page 3

We clarified that we will estimate the pelvis motion, taken as a parameter to assess balance stability, not only with the IMU, but also using the optoelectronic system - Vicon (lines 274-280 page 8).

We hope that the manuscript will be now suitable for publication in PLOS ONE.

On behalf of all authors,

Kind regards,

Maria Rubega, Ph.D.

---

## [Editor Report · Decision Letter 2]

2 Oct 2023

Experimental protocol to investigate cortical, muscular and body representation alterations in adolescents with idiopathic scoliosis

PONE-D-23-08292R2

Dear Dr. Rubega,

We’re pleased to inform you that your manuscript has been judged scientifically suitable for publication and will be formally accepted for publication once it meets all outstanding technical requirements.

Kind regards,

Andrea Tigrini, Ph.D.

Academic Editor

PLOS ONE

Additional Editor Comments: Plaase, improve the resolution of the figure you provided.

---

## [Editor Report · Acceptance letter]

5 Oct 2023

PONE-D-23-08292R2 

Experimental protocol to investigate cortical, muscular and body representation alterations in adolescents with idiopathic scoliosis 

Dear Dr. Rubega:

I'm pleased to inform you that your manuscript has been deemed suitable for publication in PLOS ONE. Congratulations! Your manuscript is now with our production department. 

Kind regards, 

on behalf of

Dr. Andrea Tigrini 

Academic Editor

PLOS ONE